# Psychosocial Distress in Adult Patients Awaiting Cancer Surgery during the COVID-19 Pandemic

David Forner [1,2,3], Sarah Murnaghan [2], Geoffrey Porter [2,4,5], Ross J. Mason [6], Paul Hong [1,2], S. Mark Taylor [1,2], James Bentley [7], Gregory Hirsch [2,8], Christopher W. Noel [3,9], Matthew H. Rigby [1,2], Martin Corsten [1,2], Jonathan R. Trites [1,2], Victoria Taylor [1], Cynthia Kendell [2], Margaret Jorgensen [2] and Robin Urquhart [2,*]

1   Division of Otolaryngology—Head & Neck Surgery, Dalhousie University, Halifax, NS B3H 2Y9, Canada; david.forner@dal.ca (D.F.); paul.hong@iwk.nshealth.ca (P.H.); smarktaylor@eastlink.ca (S.M.T.); mhrigby@dal.ca (M.H.R.); martin.corsten@nshealth.ca (M.C.); jonathan.trites@nshealth.ca (J.R.T.); victoriaataylor@eastlink.ca (V.T.)
2   Department of Surgery, Dalhousie University, Halifax, NS B3H 2Y9, Canada; sarahm.dickieson@nshealth.ca (S.M.); geoff.porter@nshealth.ca (G.P.); greg.hirsch@nshealth.ca (G.H.); cynthia.kendell@ccns.nshealth.ca (C.K.); margaret.jorgensen@ccns.nshealth.ca (M.J.)
3   Institute of Health Policy, Management and Evaluation, University of Toronto, Toronto, ON M5T 3M6, Canada; christopher.noel@mail.utoronto.ca
4   Division of General Surgery, Dalhousie University, Halifax, NS B3H 2Y9, Canada
5   Department of Community Health and Epidemiology, Dalhousie University, Halifax, NS B3H 2Y9, Canada
6   Department of Urology, Dalhousie University, Halifax, NS B3H 2Y9, Canada; rossj.mason@nshealth.ca
7   Department of Obstetrics and Gynecology, Dalhousie University, Halifax, NS B3H 2Y9, Canada; jim.bentley@dal.ca
8   Division of Cardiac Surgery, Dalhousie University, Halifax, NS B3H 2Y9, Canada
9   Department of Otolaryngology—Head & Neck Surgery, University of Toronto, Toronto, ON M5T 3M6, Canada
*   Correspondence: robin.urquhart@nshealth.ca

**Abstract:** Cancer causes substantial emotional and psychosocial distress, which may be exacerbated by delays in treatment. The COVID-19 pandemic has resulted in increased wait times for many patients with cancer. In this study, the psychosocial distress associated with waiting for cancer surgery during the pandemic was investigated. This cross-sectional, convergent mixed-methods study included patients with lower priority disease during the first wave of COVID-19 at an academic, tertiary care hospital in eastern Canada. Participants underwent semi-structured interviews and completed two questionnaires: Hospital Anxiety and Depression Scale (HADS) and Perceived Stress Scale (PSS). Qualitative analysis was completed through a thematic analysis approach, with integration achieved through triangulation. Fourteen participants were recruited, with cancer sites including thyroid, kidney, breast, prostate, and a gynecological disorder. Increased anxiety symptoms were found in 36% of patients and depressive symptoms in 14%. Similarly, 64% of patients experienced moderate or high stress. Six key themes were identified, including uncertainty, life changes, coping strategies, communication, experience, and health services. Participants discussed substantial distress associated with lifestyle changes and uncertain treatment timelines. Participants identified quality communication with their healthcare team and individualized coping strategies as being partially protective against such symptoms. Delays in surgery for patients with cancer during the COVID-19 pandemic resulted in extensive psychosocial distress. Patients may be able to mitigate these symptoms partially through various coping mechanisms and improved communication with their healthcare teams.

**Keywords:** COVID-19; cancer; psychosocial distress; waiting lists

## 1. Introduction

Cancer causes emotional and psychosocial distress as patients face disease and treatment-related mortality and morbidity, as well as grief over personal and health-

related losses, and a sense of helplessness [1]. With the COVID-19 pandemic, patients may experience further distress, owing to decreased perioperative resources and resultant fear of treatment availability. Restrictions on operating room availability for cancer patients were experienced in most Canadian jurisdictions; for varying periods of time, surgical management was limited to select, aggressive cancers. Cancer patients with less aggressive diagnoses are facing substantial treatment delays. The psychosocial distress experienced by patients who experience these delays is unknown in the literature and is hypothesized to be substantial.

Exacerbating this situation is the fact that many protective factors against patient distress are impacted by physical-distancing measures, such as diminished family support and decreased physical well-being. A more comprehensive evaluation of COVID-19 resource rationing in cancer surgery, including psychosocial outcomes, represents an important knowledge gap to consider, especially with the uncertainty of the pandemic time frame. Therefore, this study sought to examine the psychosocial outcomes of patients awaiting cancer surgery during the COVID-19 pandemic.

## 2. Materials and Methods

### 2.1. Study Design

This is a convergent, cross-sectional mixed-methods study of patients with cancer who faced treatment delays at an academic, tertiary hospital in a single payer system in eastern Canada (Halifax, Nova Scotia). Psychosocial distress was assessed at one time point using questionnaires and a semi-structured interview. This study is reported using the Good Reporting of Mixed Methods Study (GRAMS) guideline [2]. Institutional Review Board approval was obtained from the Nova Scotia Health Authority (REB#: 1025773).

### 2.2. Population

At the study institution, cancer surgery services utilize priority bands as a means to triage patients within a constrained resource environment. Four priority bands stratify patients according to expected risk over time [3]. During the first wave of the COVID-19 pandemic, beginning in March 2020, patients within the three highest priority bands received timely surgical management. We examined patients in the lowest priority band who faced delays in management.

Purposive sampling was used to select patients representative of the population of interest [4]. Representative criteria included age, sex, and cancer type. Patients were recruited from 10 June to 14 July 2020. Patients were approached by an individual within their circle of care, and subsequently by a member of the research team (SM). Participants provided written informed consent. Sample size was determined by thematic saturation, whereby participants were included until there were no newly developed themes or domains in consecutive interviews.

### 2.3. Questionnaires

Two questionnaires were administered via telephone: Hospital Anxiety and Depression Scale (HADS) and Perceived Stress Scale (PSS).

The HADS is a 14-item instrument that assesses symptoms of anxiety and depression on 21-point subscales. The HADS has shown good reliability and validity in patients with cancer [5–7]. Anxiety and depression were scored separately, and each question was scored from 0 to 3 [8]. Participants who scored from 0 to 7 were classified as within the normal range, from 8 to 10 were borderline abnormal, and from 11 to 21 were abnormal. Anxiety and depression subscales were also summed, with combined scores of 19 or higher considered to represent clinically important distress [9].

The PSS is a 10-item instrument that measures perceived stress [10]. Scores can be related to established stress levels in the general population [11]. Validity and reliability of the PSS has been established in cancer patients [12–14]. Each question was scored on a scale of 0 to 4. Scores from 0 to 13 were classified as having low perceived stress, 14 to 26

were moderate perceived stress, and 27 to 40 were high perceived stress. The mean score amongst a large U.S. population was previously identified as 13 [10].

### 2.4. Semi-Structured Interviews

Participants underwent semi-structured telephone interviews with a member of the research team (SM) using an interview guide (Supplemental Figure S1). The interview guide provided a starting point for open discussion around patient experiences, emotional well-being, and coping strategies, including communication with the treating team and changes to physical and mental health. Interviews were audio recorded and transcribed verbatim.

### 2.5. Qualitative Analysis

The qualitative portion of this study is reported in adherence to the Standards for Reporting Qualitative Research (SRQR) guidelines [15]. Thematic analysis under the Braun and Clark framework [16] with an inductive approach was used. This methodology was utilized as it allows flexibility in analysis without constraining interpretation into pre-existing theories, and allows themes to develop that are strongly linked to the data itself—in this case, the lived experiences of participants themselves. The first two interviews were independently coded (SM, RU) and used to develop a codebook. Subsequent interviews were coded, and domains were developed (SM). Themes summarizing these domains were extracted (DF, SM, RU). All steps of the qualitative analysis were performed using NVivo 12 (QSR International). Reflexivity statements may be found in Supplemental Figure S2.

### 2.6. Quantitative Analysis

Descriptive characteristics of study participants included age, sex, education level, living situation, marital status, comorbidities, prior surgical history or experiences, cancer site, and diagnosis, and whether surgery was completed prior to the interview.

### 2.7. Statistical Methods

Continuous variables were summarized as means and standard deviations or medians and interquartile ranges (IQR). Categorical variables were summarized as absolute and relative frequencies.

### 2.8. Integration

Integration of the qualitative and quantitative methods was completed using triangulation and identifying meaningful information that may have been missed or undiscovered with only one approach [17]. Questionnaire scores were compared to domains and themes at an individual participant level to ensure alignment. Results of integration are reported using joint display visualization by selection of supporting quotations in participants that experienced high (high endorsement) and low (low endorsement) levels of distress. Integration was performed by three members of the research team (DF, SM, RU).

## 3. Results

### 3.1. Participants

Fourteen of 23 (61%) eligible participants completed the study (Figure 1). The mean age of participants was 59 years (SD: 10.6, range 42–76). Most (Table 1) were female (71%), had college/undergraduate education or higher (71%), and lived with a partner in the home (79%). Most patients had a prior surgical history or experience with surgery (86%), and two-thirds had completed surgery prior to being interviewed (Table 1). The median wait time from initial consultation to time of surgery or research interview was 143 days (IQR 108–178 days).

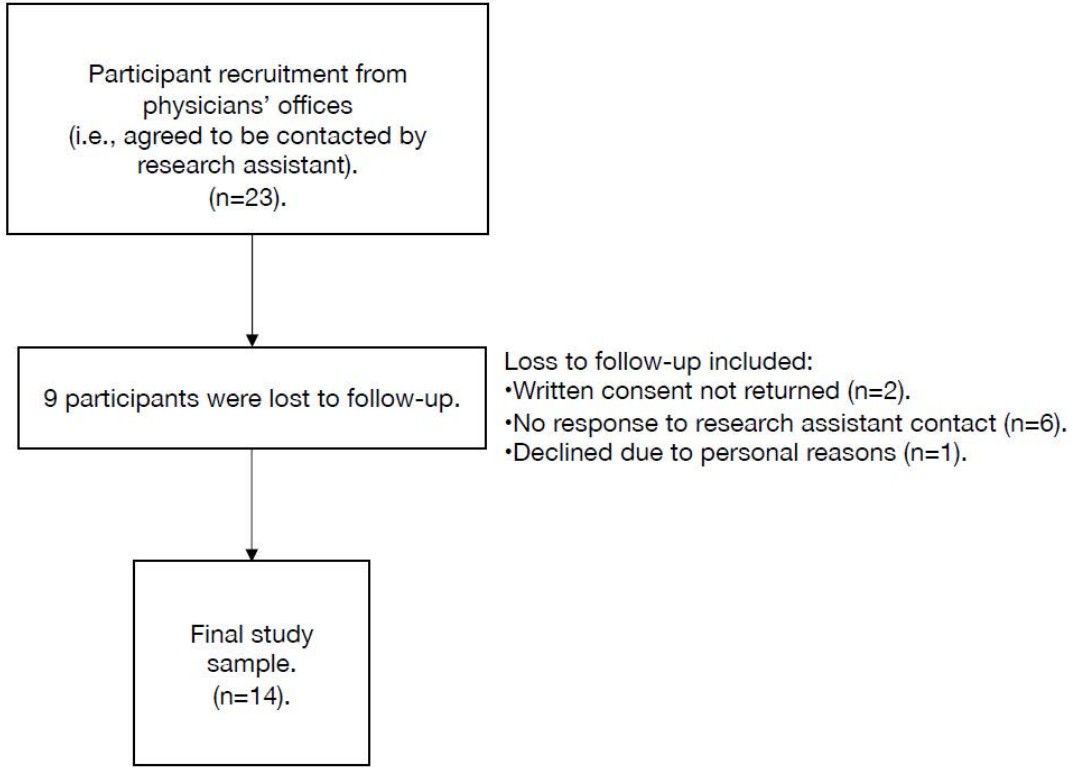

**Figure 1.** Participant flow chart.

*3.2. Qualitative Analysis*

Six themes were identified: the role of COVID-related uncertainty in fear of cancer progression (**uncertainty**), life changes as a result of the COVID-19 pandemic (**life changes**), coping strategies and how they differed from the pre-COVID era (**coping with stress**), communication quality with the healthcare team (**communication**), direct experiences with the healthcare system during the COVID-19 pandemic (**experience**), and health system decision making during the COVID-19 pandemic (**health services**). Supporting codes used throughout the qualitative analysis, as well as domains pertaining to these codes, are found in **Supplemental Table S1**.

**Table 1.** Participant Characteristics.

| Variable | n | % |
|:---:|:---:|:---:|
| Total participants | 14 | 100 |
| *Demographics* | | |
| Sex | | |
| Female | 10 | 71.4 |
| Education (highest level of completion) | | |
| Less than high school | 2 | 14.3 |
| High school | 2 | 14.3 |
| College or undergraduate university | 6 | 42.8 |
| Postgraduate university or professional program | 4 | 28.6 |
| Living situation | | |
| Children in home | 2 | 14.3 |
| Partner in home | 11 | 78.6 |
| Others in home (roommates, other family members) | 1 | 7.1 |
| Other (people other than those above) | 2 | 14.3 |
| Marital status | | |
| Married | 8 | 57.2 |
| Single/never married | 2 | 14.3 |
| Divorced/separated | 1 | 7.1 |
| Common-law | 2 | 14.3 |
| Widowed | 1 | 7.1 |
| *Clinical Characteristics* | | |
| Number of medical history diagnoses | | |
| 0 | 0 | 0.0 |
| 1 | 3 | 21.4 |
| 2 | 1 | 7.1 |
| 3+ | 10 | 71.5 |
| Prior surgical history/experiences | | |
| Yes | 12 | 85.7 |
| Disease site | | |
| Breast | 3 | 21.4 |
| Prostate | 3 | 21.4 |
| Thyroid | 3 | 21.4 |
| Kidney | 2 | 14.4 |
| Gynecological | 3 | 21.4 |
| Diagnoses | | |
| WDTC | 3 | 21.4 |
| ADH | 3 | 21.4 |
| RCC | 3 | 21.4 |
| Prostate Cancer | 3 | 21.4 |
| Uterine Cancer | 1 | 7.1 |
| Endometrial Adenocarcinoma | 1 | 7.1 |
| Completed surgery prior to interview | | |
| Yes | 9 | 64.3 |

### 3.2.1. Uncertainty

The effects of COVID-19 on the healthcare system resulted in substantial uncertainty for participants. Upon hearing that their surgeries were delayed, participants had no knowledge of when their surgery would take place or when they would hear from their

surgeons' offices. For most participants, this uncertainty resulted in feelings of distress, as well as high levels of fear of cancer progression while waiting for their surgery. As discussed by one participant, the waiting:

> " . . . was an emotional rollercoaster basically. It's something I never want to experience again because I'm not a doctor, all I knew that I have a tumor that had to be removed, and waiting. . . . I had to wait six months for my surgery because of COVID. And so me not knowing, that's a lot to worry because I don't know if it's growing or spreading. . . . On a daily basis, that was very stressful. And it was really hard to concentrate because it couldn't leave your mind, really." [Participant #9]

Several participants also described fears of increased complications or long-term effects due to their delay. For example, one prostate cancer patient stated:

> "The prospect of this expanding into colon cancer was not a nice thought. So I had extreme concerns about that. As you know, I mean, one of the other issues about this, of course, is being prostate[M1] cancer, is incontinence, erectile dysfunction, sexual function. These are all very emotional and difficult life changing possibilities. And of course, with the delay, you know, how much worse could these things go? Would this guarantee that I would be incontinent, that I would have to wear diapers? You know, these are highly amplified by the delay." [Participant #13]

Similarly, participants expressed a desire to proceed with their surgery as quickly as possible as a means to "get it [the cancer] out." As described by one participant:

> "Whether or not the COVID-19 is here, whether the COVID-19 is going to strike me, get rid of this cancer. Get it out. And this is why I think . . . Like yes, I understand everything about the COVID-19. But when there's a major surgery that has to be done, those hospitals should automatically put us through. . . . I could have died waiting for that surgery. Not from the COVID-19 but what was in my body . . . " [Participant #11]

### 3.2.2. Life Changes

Participants described how their heightened distress was further amplified by lifestyle-related changes associated with COVID-19. That is, participants discussed how the effects of social isolation and physical distancing, and alterations to their daily lives (for example, loss of income, needing to work from home, home schooling, and the inability to partake in usual activities and hobbies), impacted their health and wellbeing while waiting for their cancer surgery. One participant described his surgical delay alongside a loss of income during COVID-19 as "a double whammy stress" [Participant #13].

At the same time, most participants felt they did well in spite of the many changes that the COVID-19 pandemic brought to their lives. As stated by one participant:

> "I'm an obsessive gardener and reader and knitter. And so there hasn't been that much difference in that respect. . . . it was very annoying because I like to walk, and I couldn't go to the beaches and so on and so forth. But I managed. I can be pretty creative when necessary. So compared to many people, I'm all right." [Participant #19]

### 3.2.3. Coping with Stress

Most participants discussed trying to maintain a routine and a sense of normalcy (for example, by keeping busy, staying active, and taking up hobbies) as ways of coping during their surgical delay.

While participants developed coping strategies, their feelings of distress were only partially mitigated, with many still experiencing substantial amounts of distress. For example, one participant described:

"I guess when I get stressed, I probably kind of close down or try not to think about it. I've been trying to keep busy doing other things so I don't think about it. But then it gets to a point that, you know, you just . . . It's hard." [Participant #7]

Family, friends, and various community groups (for example, church) were also seen as important sources of support to mitigate their distress and help cope during the delay.

"I mean we have a lot of support. We have a lot of family and friends and church people that they all know. Like, I let everybody know that this is what I have to have done, and the date of my surgery. And everybody is very supportive and have sent a lot of encouraging messages and things to me." [Participant #5]

However, some participants also discussed shielding their loved ones from their diagnosis and their added distress related to their surgical delay. Common reasons for this were not wanting to increase the burden placed on their family due to COVID-19 and not wanting to cause additional stress with uncertainty related to their cancer. One participant said:

"Well, the stress and the strain that I was under . . . Like my [family member] had a stroke two years ago, and I like to keep things away from him that's going to get him stressed out and that. So, I'm doing everything I can not to like talk about it." [Participant #11]

3.2.4. Communication

The quality of one's communication with one's healthcare team was commonly discussed. For those who had opportunities to communicate with their surgeon or oncologist during the delay, they described this communication as reducing their distress and allaying their fears related to the delay. As one participant described:

"I'd say that the impact and the anxiety was all before I had enough information. Once I had enough information . . . And you know, hats off and great thank you to both of those specialists that I mentioned at how communicative they were and how not one phone call that I had with them felt like I was taking up their time or that there was a time limit on the call. So what got me through and eased the anxiety and everything was good information in the meantime" [Participant #14]

Conversely, many participants described experiences where they received almost no contact or communication from their specialists, with some even unaware of their surgery being delayed. Several participants also discussed actively trying to contact their surgeons' offices and receiving only voice mail recordings and pre-recorded messages, without receiving a follow-up phone call. As described by one participant:

"And of course, every time I called, I would just get a recording that said, 'Everything's shut down; everything's shut down'. . . . It was stressful . . . very stressful." [Participant #15]

This lack of communication led to challenges in navigating the healthcare system during their treatment course. Participants suggested that a centralized contact for COVID-19 and surgical waitlist information would have been invaluable to easing their experience:

"I just wish sometimes that I could have . . . if there was a number that you could have called to just say, 'Hey, what's going on or do you think surgeries would be opening up soon?'" [Participant #18]

Participants who had poor experiences with communication highlighted their continual frustration, distress, and fear of disease progression:

"Like I was frustrated because I mean I would call Dr. [X's] office and not really get any kind of response, you know? Like you'd get a message about COVID and appointments are . . . there's no appointments happening, you know, because of COVID, and we'll contact you. And you'd leave messages and . . . You know, I left a couple of messages and never got any response, you know?" [Participant #7]

### 3.2.5. Experiences

A minority of patients were concerned about the risk of catching COVID-19 through their access to the healthcare system (for example, entering the hospital for surgery):

"My concern was, and maybe I was selfish in thinking this, oh, my gosh, if I get to the hospital . . . if they do my surgery during, you know, when COVID first started, if I did go in for my surgery, how will I know . . . what if other nurses and doctors who were working with people all the time, what if they're around somebody that has COVID and didn't know it, and then they get it, and they're asymptomatic so they won't know? And what if I pick it up, and I'm so not strong, gosh, would I be able to fight COVID along with everything else?" [Participant #18]

Conversely, those patients who had already undergone surgery at the time of their interview voiced feeling positive and reassured with their direct, first-hand experiences with the health care system:

" . . . The people at the hospital, the staff, everybody that was there—everybody— made you realize, like, you know, you're so lucky that . . . You know, I had to wait but I was so lucky to be in good hands afterwards. Like they were amazing. Like I don't know how they do these jobs, but they were all amazing, which made me feel so much better." [Participant #18]

### 3.2.6. Health Services

Most participants understood the reasons for the surgical delays and hospital-related restrictions, despite being concerned about their own health and cancer progression and finding it difficult to be in hospital without family. At the same time, many questioned decisions around health-services delivery during the COVID-19 pandemic, including the cancellation or delays of cancer surgeries. There was a strong expression that the health system must maintain an ability to provide care for ill patients during a pandemic and not delay vital care to patients with potentially life-threatening conditions. Participants described their views this way:

" . . . one wonders . . . what's the best evil here? Clearly, the public good is more important. But at what point is what potentially is I guess it's determined to be not life threatening, but at what point is prostate cancer less and less important than other life-threatening situations?" [Participant #13]

### 3.3. Quantitative Analysis

The majority of patients had borderline abnormal, or abnormal, anxiety symptoms (Table 2), including 36% with abnormal anxiety and 14% with abnormal depression. Almost half of the participants had borderline abnormal or worse depressive symptoms (Table 2, 3%). Five (36%) patients had total HADS scores of 19 or greater, suggesting clinically important levels of anxiety and depression.

Similarly, participants experienced a high degree of stress, with 64% experiencing at least moderate stress due to delays in their cancer care (Table 2), including 14% with high levels of stress. The median stress score on the PSS questionnaire was 19.0 (IQR 11.5).

In a post-hoc, unadjusted analysis, there was a non-significant difference in mean reported distress between participants who had undergone surgery and those who had not, including anxiety symptoms (7.8 vs. 8.4, $p = 0.61$), depressive symptoms (4.5 vs. 6, $p = 0.59$), and perceived stress (17.3 vs. 17.9, $p = 0.89$).

**Table 2.** Summary of psychosocial distress questionnaire scores.

| Questionnaire | Score Range | N | % | Median | IQR |
|---|---|---|---|---|---|
| *HADS* | | | | | |
| Total Score | 0–42 | 14 | 100 | 12.5 | 14.3 |
| Anxiety score | 0–21 | 14 | 100 | | |
| Normal | 0–7 | 5 | 35.7 | | |
| Borderline abnormal | 8–10 | 4 | 28.6 | 8.5 | 6.5 |
| Abnormal | 11–21 | 5 | 35.7 | | |
| Depression score | 0–21 | 14 | 100 | | |
| Normal | 0–7 | 8 | 57.1 | | |
| Borderline abnormal | 8–10 | 4 | 28.6 | 5.0 | 7.8 |
| Abnormal | 11–21 | 2 | 14.3 | | |
| *PSS* | | | | | |
| Total score | 0–40 | 14 | 100 | | |
| Low stress | 0–13 | 5 | 35.7 | | |
| Moderate stress | 14–26 | 7 | 50.0 | 19.0 | 11.5 |
| High perceived stress | 27–40 | 2 | 14.3 | | |

*3.4. Integration*

Joint display visualization of the qualitative and quantitative results is presented in Table 3. Themes revealed through qualitative analysis aligned well with both individual-level questionnaire scores as well as median scores, suggesting high triangulation between the two methods of investigation. Participants described substantial distress associated with delays in their cancer surgery, and this was reflected with elevated symptoms of anxiety, depression, and stress.

**Table 3.** Integration of quantitative and qualitative findings through joint display.

| Questionnaire | Score (Median, IQR) | High Endorsement of Symptoms Score | Supporting Quote | Low Endorsement of Symptoms Score | Supporting Quote |
|---|---|---|---|---|---|
| HADS | 12.5 (14.3) | 26 | Oh, I could not sleep. I did not sleep, no. It was to the point where I had to concentrate and start trying a routine to figure out how to sleep because I was just too worried. | 4 | You know, I've wondered, you know, is that going to continue to spread or how fast might that spread or will they find more when they do surgery? So I have had that thought. But I certainly realize there's nothing I can do about it. What is, is. What's there is there. But no, I'm not, you know, depressed or anything like that. |
| PSS | 19.0 (11.5) | 32 | Well, it was extremely stressful being told that everything was cancelled. That was very frightening to me. It's like what do you mean cancelled? | 5 | I mean no one's looking forward to surgery in the first place. But because they never gave me a sense of urgency in the first place, that I felt, you know, okay, about it. |

**4. Discussion**

In this study of patients facing prolonged wait times for cancer surgery during the first wave of the COVID-19 pandemic, participants were found to have substantial levels of psychosocial distress. Coping strategies and excellent communication with their healthcare team were able to mitigate some of these feelings. Nonetheless, symptoms of anxiety and depression, perceived stress, and fear of their cancer progressing remained high.

Distress amongst cancer patients was previously suggested to be a sixth vital sign by several national cancer societies [18,19]. Amongst patients with newly diagnosed cancer, anxiety associated with waiting for treatment is exceptionally common [20]. In patients with thyroid cancer, stress, anxiety, and intrusive thoughts are pervasive for those awaiting thyroidectomy [13]. Concomitantly, cancer patients experience a significantly decreased quality of life while awaiting surgery, much of which is attributed to mental health impairment [21]. Recently, Gagliardi and colleagues explored the available literature on the psychosocial impact of waiting for surgery and highlighted the need for studies

specifically examining the additional distress experienced by patients awaiting surgery during the COVID-19 pandemic [22].

Participants in this study reported distress as a result of both their cancer diagnosis and the uncertainty associated with the COVID-19 pandemic. However, some individuals were able to reduce this distress partially by instituting various coping strategies. Interestingly, several participants also highlighted both the importance of communication with the healthcare team and the perceived benefit of a centralized system for obtaining surgery-related COVID-19 information. The use of centralized referral and triage systems for oncology services during COVID-19 was already proposed [23]. Integration of patient update platforms, including wait times and changes to service availability, may be a straightforward means to alleviate distress for patients waiting for treatment and may reduce the burden of providing these updates for individual physician offices.

The findings of this study must be interpreted within the context of its design. Participants in this study were recruited from a universal healthcare system where prioritization of healthcare services are guided by a collection of factors, including the expected severity of a diagnosis. Thus, time to surgery is not typically at the sole discretion of the patient or surgeon themselves. In this study, the median wait to surgery was reflective of the pandemic in our jurisdiction, which experienced a lower per-capita rate of COVID-19 infections compared to other regions in Canada and worldwide. Nonetheless, participants and healthcare providers could not predict the length of health system restrictions while actively waiting for surgery. While steps were taken to enhance the rigor of the analysis, including development of a codebook by two team members and team debriefing, implicit experiences may be present from the perspective of the research team, in which all members are either researchers or clinicians interested in the care of patients with cancer. Some potential participants who expressed interest in the study were lost to follow-up and unable to be included in the study. Such individuals may have experienced psychosocial distress different from that expressed by participants. Participants who underwent surgery before study completion may have reported less wait-time-related distress. In a post-hoc analysis, reported anxiety-, depressive-, and stress-related symptom scores were nonsignificantly lower in patients who had undergone their operative procedure before the time of study participation. This study was not designed, nor powered, to explore this relationship definitively. However, despite including such patients, the findings of this study still support additional distress associated with pandemic-associated delays.

As questionnaire scores for a hypothetical control cohort of patients without a COVID-19-related delay during the same time period are inherently unavailable, it is not possible to determine the effect size of COVID-19 on psychosocial distress. Alternative groups, such as those with high-priority diagnoses not experiencing delays, or benign conditions not experiencing the distress of cancer, represent different patient populations and therefore were not applicable for comparison. However, triangulation does support additional stress beyond the underlying cancer diagnosis, suggesting at least some degree of increased psychosocial distress in the setting of the pandemic. Indeed, compared to literature values of patients with similar low-risk cancers, participants in this study had higher HADS and PSS scores [24–26].

This study represents the first to investigate the psychosocial distress experienced by cancer patients facing prolonged wait times during the COVID-19 pandemic. Currently, the COVID-19 pandemic remains a significant issue throughout the world, with many locations experiencing either resurgent waves or continued first wave infections. The findings of this study highlight the need for additional supports as well as possible solutions to minimize distress experienced by our patients. Future research regarding how to manage and reduce the distress of these patients is important, including the use of centralized reporting and prioritization systems, improvement of access to support groups, and identification of other widely applicable interventions. These findings may be applicable to potential future global events, and indeed to cancer patients facing routine wait times not impacted by such catastrophic health system stressors.

## 5. Conclusions

Delays in surgery for patients with cancer during the COVID-19 pandemic resulted in psychosocial distress. Patients may be able to mitigate these symptoms partially through various coping mechanisms. The addition of centralized systems for the reporting of healthcare service delays may be beneficial for patients and may help further reduce distress.

**Supplementary Materials:** The following are available online at https://www.mdpi.com/article/10.3390/curroncol28030173/s1, Table S1: Supporting codes used throughout the qualitative analysis with accompanying domains.

**Author Contributions:** Conceptualization, D.F., S.M., G.P., R.J.M., P.H., S.M.T., G.H., C.W.N., J.B., M.C., J.R.T., C.K., M.J., R.U.; methodology, D.F., S.M., C.W.N., V.T., P.H., M.H.R., M.C., C.K., R.U.; formal analysis, D.F., S.M., R.U.; investigation, D.F., S.M., R.U.; resources, G.P., G.H., R.U.; data curation, S.M.; writing—original draft preparation, D.F., S.M., R.U.; writing—review and editing, all authors; visualization, D.F., S.M., R.U.; supervision, R.U.; project administration, S.M., C.K., M.J.; funding acquisition, G.P., R.U. All authors have read and agreed to the published version of the manuscript.

**Funding:** Operational funding was provided through the Ramia Chair in Surgical Oncology held by Geoffrey Porter. David Forner received graduate student funding through the Dalhousie University Faculty of Medicine Killam Scholarship and the Ross Stewart Smith Fellowship in Medical Research.

**Institutional Review Board Statement:** The study was conducted according to the guidelines of the Declaration of Helsinki, and approved by the Institutional Review Board of Nova Scotia Health Authority (protocol 1025773 and 11 2020 May).

**Informed Consent Statement:** Informed consent was obtained from all subjects involved in the study.

**Data Availability Statement:** The data presented in this study are available on request from the corresponding author. The data are not publicly available due to privacy and confidentiality considerations.

**Conflicts of Interest:** The authors have no conflict of interest to disclose.

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
