# Peer review of "Psychosocial Distress in Adult Patients Awaiting Cancer Surgery during the COVID-19 Pandemic"

_curroncol, doi:10.3390/curroncol28030173_

Round 1
Reviewer 1 Report
Thank you for the opportunity to review this timely manuscript on an important topic – the psychosocial impact of waiting for procedures that were cancelled or delayed due to the focus of healthcare resources on the pandemic.
Background
Concise, clear and to the point. Identifies gap in knowledge: while considerable research has studied the logistics of wait times, little prior research has examined the psychosocial impact of waiting for procedures. Such knowledge is needed to identify strategies that may be needed to reduce anxiety, stress, depression, etc. This is particularly true for patients already experiencing distress due to a cancer diagnosis.
Methods
Mixed methods is an appropriate design and the study complied with GRAMMs reporting criteria. Eligibility, sampling and recruitment strategies are described. Instruments are well-described. The qualitative component complied with SRQR reporting criteria. Analysis for both the quant and qual components, and integration of knowledge from both components is sufficiently addressed.
Suggestions for Methods:
- Mention if there was a target sample size, why, and how final sample size was determined.
- Specify the qualitative approach used (e.g. basic qualitative description?) and why that approach was used
- Although the interview guide is available in a supplemental file, provide a sentence specify the key questions so that readers understand how the interview component supplemented data gathered by instruments
Results
Participant characteristics and loss to follow-up are well described. Themes and instrument scores are well presented in table and text with select quotes. Table 3 does a good job of displaying how quant and qual findings are complementary
Suggestions for Results:
- Figure 1 could be a little better executed graphically-speaking
- I like that quotes are sprinkled through the Results but they would be easier to read if they stood out from the text; consider adding a blank line before and after each quote and indenting them?
- Can you contrast the 2/3 who had had their surgery at the time of participant with the 1/3 who had not yet had their surgery? One would expect higher distress in the latter group
Discussion/Conclusions
Appropriately contextualizes the findings, noting strengths and limitations, and recommending additional research on strategies to address the psychosocial impact of waiting.
Suggestion for Discussion:
You might reference the following in the Discussion, only to emphasize that there is little prior research on the psychosocial impact of waiting or strategies to combat it, and that this study specifically in the pandemic context is an important addition to the literature: Gagliardi AR et al. The psychological burden of waiting for procedures and patient‐centred strategies that could support the mental health of wait‐listed patients and caregivers during the COVID‐19 pandemic: A scoping review. Health Expectations 2021; https://doi.org/10.1111/hex.13241
Reviewer 2 Report
Dear Authors,
Psychosocial distress in adult patients awaiting cancer surgery during the COVID-19 pandemic - it is a very important topic nowadays.
For affiliations - the country of authors should be given.
The research design is appropriate. Material and methods, results and discussion are adequately described. The results are clearly presented.
line 91: 2.4. Semi-Structured Interviews - this part should be pass the sentence to the next page.
Figure 1 should be completly on page 4.
Shame 3 - Please standardize the bullet points.
line 323 - this part should be pass the sentence to the next page.
References should be rewritten according to the guidelines for authors.
To sum up, this article can be accepted after corrections.
